# Effectivity of (Personalized) Cognitive Behavioral Therapy for Insomnia in Mental Health Populations and the Elderly: An Overview

**DOI:** 10.3390/jpm12071070

**Published:** 2022-06-29

**Authors:** Teus Mijnster, Gretha J. Boersma, Esther Meijer, Marike Lancel

**Affiliations:** 1Centre of Expertise on Sleep and Psychiatry, GGZ Drenthe, Mental Health Institute, 9404 LA Assen, The Netherlands; teus.mijnster@ggzdrenthe.nl (T.M.); gretha.boersma@ggzdrenthe.nl (G.J.B.); esther.meijer@ggzdrenthe.nl (E.M.); 2Forensic Psychiatric Hospital, GGZ Drenthe, Mental Health Institute, 9404 LA Assen, The Netherlands; 3Department of Clinical Psychology and Experimental Psychopathology, University of Groningen, 9712 TS Groningen, The Netherlands

**Keywords:** mental health disorder, elderly, insomnia, cognitive behavioral therapy for insomnia, treatment adaptations

## Abstract

Insomnia is very prevalent in psychiatry and is considered a transdiagnostic symptom of mental disorders. Yet, it is not only a consequence of a mental condition but may also exert detrimental effects on psychiatric symptom severity and therapeutic response; thus, adequate insomnia treatment is particularly important in psychiatric populations. The first choice of intervention is cognitive behavioral therapy for insomnia (CBT-I) as it is rather effective, also in the long run without side effects. It is offered in various forms, ranging from in-person therapy to internet-delivered applications. CBT-I protocols are typically developed for individuals with insomnia disorder without co-occurring conditions. For an optimal therapeutic outcome of CBT-I in individuals with comorbid mental disorders, adaptations of the protocol to tailor the treatment might be beneficial. Based on a literature search using major search engines (Embase; Medline; APA Psych Info; and Cochrane Reviews), this paper provides an overview of the effectiveness of the different CBT-I applications in individuals with diverse comorbid mental conditions and older adults and describes the functionality of CBT-I protocols that have been personalized to specific psychiatric populations, such as depression, substance abuse, and schizophrenia spectrum disorder. Finally, we discuss urgent needs for insomnia therapy targeted to improve both sleep and psychopathologies.

## 1. Introduction

Insomnia disorder has adverse effects on a broad range of cognitive processes, such as attention, memory, and executive functioning (e.g., [1,2]). Furthermore, short and disrupted sleep negatively impacts physical health, for example, by increasing the risk of cardiovascular, metabolic, and immune disorders [3,4,5,6]. There is also increasing evidence that insomnia has detrimental effects on emotion regulation and mental health. Even in otherwise mentally healthy persons, having insomnia leads to irritability, moodiness, and negative affect [7], and increases the risk of suicidal thoughts and behavior [8]. For those with a mental disorder, insomnia may further increase the severity of their mental health symptoms and impede treatment response and recovery prospects (e.g., [9,10]). As insomnia is very common in mental health populations, early diagnosis and treatment of comorbid insomnia disorder is paramount for psychiatric outcome.

In psychiatric practice, treatment of insomnia disorder generally consists of pharmacotherapy, e.g., benzodiazepines or low doses of sedative psychopharmaceuticals, which temporarily promote sleep but do not cure insomnia disorder. Worldwide, the first-choice treatment for insomnia is cognitive behavioral therapy for insomnia (CBT-I) [11,12]. This therapy, which stimulates physiological sleep, is highly effective [13,14], also in the long run [15], with minimal adverse effects, and without the risk for tolerance and physical and mental dependency. Historically, CBT-I was developed for primary insomnia and was discouraged in persons with comorbid disorders. However, over the last two or three decades, growing evidence suggests that CBT-I has positive outcomes for mental health populations as well (e.g., [16,17,18]). Although CBT-I is effective, access is often limited, especially due to high treatment intensity and lack of trained professionals. To improve the accessibility of this intervention, shorter versions of CBT-I protocols with optional modules have emerged. Alongside, different delivery methods, such as group, self-help, or internet-delivered CBT-I, are becoming increasingly available. In mental health patients, characteristics related to their specific mental disorder may hamper feasibility, treatment engagement and compliance, and overall therapeutic potential. The addition of mental disorder specific components to CBT-I may enhance the therapeutic benefit for both sleep and co-occurring psychopathology. Therefore, several adaptations to the standard CBT-I protocol to personalize the insomnia treatment to a patient’s mental health characteristics have been developed. 

In this narrative review, we aim to describe existing modifications of CBT-I and their efficacy in specific mental health populations. First, healthy sleep, sleep assessment, insomnia disorder, and the most common insomnia treatments are discussed. Hereafter, a description is given of CBT-I and related non-pharmacological insomnia therapies for the following psychiatric disorders: major depressive disorder (MDD), bipolar disorders, anxiety disorders, posttraumatic stress disorders (PTSD), substance use disorders (SUD), schizophrenia spectrum disorders (SSD), attention deficit hyperactivity disorders (ADHD), and autism spectrum disorders (ASD). Additionally, specific attention is paid to insomnia in elderly persons. Finally, an overarching conclusion, gaps in the literature, and future perspectives are provided.

## 2. Methods

Scientific papers on the evaluation of cognitive and/or behavioral therapy for insomnia in specific mental health populations were obtained using common search engines (PubMed, APA PsychINFO, Embase, Google Scholar, Cochrane). Furthermore, relevant references were checked and included. Only papers describing sleep outcomes and written in English, Dutch, or German were considered. Initial search terms included: cognitive, behav*, therapy, treatment, intervention, insomnia, adaptation, modification, tailored, psychiatric/mental disorder/disease, depress*, bipolar, anxiety, ADHD, PTSD, autism, schizophrenia, psychosis, substance use/abuse/dependency, alcohol, opioids, stimulants, cannabi*, amphetamine, elderly, older adults, geriatric. For description, we preferred articles on studies with a minimum sample size of 20 and a control condition. All papers discussed in this review are briefly described in Table 1.

## 3. Sleep and Insomnia Disorder

### 3.1. Neurophysiology of Sleep

Sleep is characterized by a reversible altered state of consciousness and a reduced ability to react to external stimuli. It consists of two distinct stages: rapid eye movement (REM) sleep and non-REM sleep, which is subdivided into three stages of increasing sleep intensity (N1, N2 and N3). The gold standard method to assess sleep is polysomnography (PSG): the simultaneous recording of brainwave activity (electroencephalography (EEG)), eye movements (electrooculography) and skeletal muscle activity (electromyography). Non-REM sleep is characterized by synchronized EEG signals with a low frequency, such as sleep spindles during N2 and slow waves during N3, ‘deep sleep’, and gradual reduction of muscle activity from N1 to N3 [57]. REM sleep is associated with desynchronized fast EEG waves, rapid eye movements, and muscle atonia. As most dreams occur during REM sleep, this sleep stage is often called dream sleep. Non-REM and REM sleep alternate and in adults form non-REM–REM cycles of approximately 90 min occurring four to six times in a night’s sleep. Deep sleep is mainly present during the first part of the night, while N2 and REM sleep dominate the second half of the night.

There are large inter-individual differences in the quantity and quality of sleep (e.g., [58]). Yet, the following values are generally assumed ‘normal’ in adults: sleep latency (time between lights out and onset of sleep) below 30 min, total sleep time 7–9 h, sleep efficiency (% sleep of time in bed) above 85%, number of awakenings below five, wakefulness after sleep onset below 30 min, and N1 1–5%, N2 50%, N3 20–25%, and REM sleep 20–25% of total sleep time. A rule of thumb is that a person obtains the right amount of sleep when he/she wakes up feeling well rested, functions well during the day, and is able to fall asleep at their habitual bedtime. Sleep varies across lifespan: aging is associated with a reduction of deep sleep and REM sleep and a decrease of sleep efficiency due to more frequent and longer lasting arousals and awakenings (e.g., [59,60]).

The most widely applied model of sleep–wake regulation is the two-process model, which posits that sleep–wake behavior is governed by two distinct and separate processes [61]. The first is process S (sleep pressure), a homeostatic process that builds up during wakefulness, thereby creating a sleep drive, and declines during sleep. The second, process C (circadian rhythm), has the form of a sine wave and is controlled by the twenty-four-hour circadian clock located in the suprachiasmatic nucleus. Process C, which promotes alertness, starts to rise in advance to awakening, continues to rise throughout the day, and declines from a few hours before bedtime onwards. For humans, the night is the optimal time period to sleep, as sleep pressure is high due to preceding wakefulness, and the circadian drive for wakefulness is low. When aligned, process S and process C enable us to sleep for 8 h and be awake and alert for 16 h at a stretch.

Apart from the influence of the circadian clock, the induction and maintenance of sleep is mostly regulated by the hypothalamus and the brain stem. In the hypothalamus, the simultaneous release of the inhibiting neurotransmitter GABA [62] and the inhibition of the activating neurotransmitter orexin [63,64] inhibit aminergic neurotransmission (histamine, dopamine, serotonin, noradrenalin, and acetylcholine) [65,66]. This leads to a synchronization of neurons in the thalamus and cortex as evidenced by sleep spindles and slow waves that typically occur during non-REM sleep [57]. In contrast to non-REM sleep, acetylcholine neurons are active during REM sleep, and this may underlie the wake-like desynchronized neuronal activity observed in REM sleep [67]. Further, signals from the brainstem inhibit somatic motor neurons, resulting in the lack of motor tone (movement) during REM sleep [68,69].

### 3.2. Insomnia Disorder

Insomnia symptoms consist of problems falling asleep, staying asleep, and early morning awakening. Acute or short-term insomnia lasts several days or weeks and is generally caused by physical or psychological stressful events. It may disappear upon improvement of the trigger, but often evolves into a chronic condition [70]. Insomnia disorder is defined as frequent (≥3 nights per week) and long-lasting (≥3 months) difficulties initiating and/or maintaining sleep, causing significant daytime distress/impairments, such as fatigue, attention, and memory problems, and is not better explained by other sleep disorders (*International Classification of Sleep Disorders* third ed.; ICSD-3 [71]) or physical or mental illnesses (DSM-5 [72]). Acute insomnia is generally thought to be due to predisposing (vulnerability) and precipitating (triggering) factors, whereas its progression to an insomnia disorder is induced by perpetuating (maintaining) factors. The latter comprises behaviors, such as lying in bed for long time periods and taking daytime naps, and cognitive processes, such as fear of sleeplessness and its daytime consequences, as well as the resulting mental and physical hyperarousal [73,74]. The perpetuating factors render chronic insomnia that is independent of its original cause. Insomnia is a rather common condition: 33–50% of the general population experience insomnia symptoms and about 10% suffer from insomnia disorder [75,76]. Risk factors for insomnia disorder include older age, female sex, (chronic) medical diseases, mental disorders, and other sleep disorders [77]. 

### 3.3. Insomnia and Mental Health

Estimates of the prevalence of insomnia in psychiatric populations vary greatly between studies and are dependent on psychopathology, definition, and method of assessment of insomnia. However, the prevalence of insomnia reported in individuals with a psychiatric disorder, ranging from 15 to 60% [78,79,80], seem consistently higher than those reported in the general population [81,82,83]. This high prevalence is of particular importance as an increasing number of studies show strong associations between insomnia and mental health symptom severity and insomnia being a predictor for future mental disorders (e.g., [9,84]). Thus, early identification of insomnia could be beneficial for the prevention of and the recovery from mental disorders. However, the identification and treatment of insomnia disorder in psychiatric patients is often complicated by the view of insomnia as a symptom of psychopathology that will resolve when the patients recover from the psychiatric disorder; yet, insomnia often persists after remission [85,86] and poses a relapse risk [87,88]. Furthermore, treatment of insomnia has been shown to not only improve sleep but also to ameliorate symptoms of the comorbid psychiatric illness [31,89,90]. 

### 3.4. Sleep Assessment

To diagnose insomnia disorder, an accurate clinical assessment according to ICSD-3 [71] of, amongst others, sleep history, present sleep quality, and sleep–wake behavior is indicated. Furthermore, sleep questionnaires can be employed to measure subjective/experienced sleep. The most frequently used questionnaires in sleep research are the Insomnia Severity Index (ISI) and the Pittsburgh Sleep Quality Index (PSQI). The ISI consists of seven items concerning sleep problems rated on a five-point Likert scale (0 = no problem; 4 = severe problem). The total score (0–28) reflects insomnia severity and is often used to determine whether a treatment response has occurred (ISI decreases ≥ 8 points from pre- to posttreatment) and/or insomnia has remitted (ISI measured posttreatment either <7 or <8). The ISI is a reliable and valid instrument [91]. The PSQI is a 19-item self-report questionnaire. Items, ranging from 0 (no difficulty) to 3 (severe difficulty), are grouped into seven component scores and are summed to generate the global score (0–21), with higher scores reflecting poorer subjective sleep quality. Global scores above five indicate poor sleep. This instrument has good psychometric properties [92]. Sleep diaries are often used in clinical practice and trials to assess sleep prospectively for longer time periods. Examples of standardized sleep diaries are the Pittsburgh Sleep Diary [93] and the Core Consensus Sleep Diary [94].

Although objective sleep assessments are not required for the diagnosis of insomnia disorder, they are regularly employed in sleep clinics to diagnose or exclude other (comorbid) sleep disorders such as obstructive sleep apnea syndrome (OSAS). Nocturnal PSG delivers a broad range of sleep variables ranging from sleep latency and amount of N3 to the number of awakenings. However, it is a very time-consuming, laborious, and expensive measurement, mainly used when other severe sleep disorders are suspected. A more feasible and less-costly method is actigraphy: the monitoring of motion by a device worn around the non-dominant wrist, whereby periods of inactivity are interpreted as sleep. When worn for a longer time period, preferably several weeks, actigraphy provides information on variables such as sleep–wake timing, sleep latency, and total sleep time.

### 3.5. Insomnia Treatment

Pharmacological treatment: To treat insomnia, hypnotics are usually prescribed; benzodiazepines and Z-drugs effectively improve both sleep onset and sleep maintenance [95,96]. However, remission of insomnia in real-world settings is often not achieved, especially when psychiatric comorbidities are present [97]. Moreover, the drugs have highly undesirable effects such as daytime fatigue or somnolence, amnesia, muscle weakness, development of tolerance, dependence, and rebound insomnia upon abrupt discontinuation [95]. Due to these negative effects, their use should be limited to a short time period. Off-label medication often prescribed for insomnia includes various sedative antidepressants and second-generation antipsychotics. However, they are associated with adverse (metabolic) effects and evidence for their efficacy in treating insomnia disorder is scarce [12,98]. Furthermore, they may cause or exacerbate other sleep disorders, such as restless legs syndrome and nightmare disorder [99]. Therefore, their use is only advisable when one also wants to utilize their effects for a comorbid psychiatric disorder. Because sleep medication has limited long-term benefits and is associated with negative effects on sleep (reduction of N3 and REM sleep) and daytime functioning, it is not the first choice when treating insomnia. However, pharmacological treatment can be a useful addition to behavioral and psychotherapeutic interventions for insomnia. In addition, it can be an alternative approach, for instance, when non-pharmacological treatment is not available, unsuitable (inappropriate) for a specific patient, or the treatment response is insufficient.

Cognitive behavioral treatment for insomnia: The gold standard treatment for insomnia is CBT-I, which directly targets the insomnia perpetuating factors that transition acute insomnia into chronic insomnia. Standard protocol CBT-I consists of up to six sessions, with two additional follow-up sessions, delivered by a trained clinician to patients individually or in a group (e.g., [100]). In addition to sleep education, sleep hygiene, and relaxation techniques, important components of CBT-I are the behavioral interventions sleep restriction and stimulus control, and cognitive therapy. In cognitive therapy, maladaptive beliefs and attitudes about sleep are identified and addressed. In sleep restriction, sleep pressure is increased by limiting time in bed to sleep diary-based total sleep time plus 30 min (time in bed >6 h and wake-up time should be stable). Due to the resulting sleep debt, sleep efficiency will improve and time in bed can be increased gradually. Stimulus control aims to strengthen the association between bed and sleep and to counteract conditioned arousal, amongst others, by using the bedroom only for sleep and sex, only going to bed when sleepy, and, when unable to sleep, leaving the bedroom and performing a quiet distracting activity. CBT-I exerts large positive and well-sustained effects on several insomnia symptoms both in cases of primary and comorbid insomnia (e.g., [14,16]).

Brief behavioral treatment for insomnia: Brief behavioral treatment for insomnia (BBT-I) was developed to meet some of the limitations of standard CBT-I such as shortage of trained clinicians, long treatment duration (6–9 weeks), and high treatment intensity [101,102]. It is a short intervention (two in-person sessions and two brief telephone sessions) that is focused on sleep education, sleep restriction, and stimulus control, and can be delivered by primary care nurses and is acceptable to patients [52,103]). It has been shown to effectively improve sleep with moderate to large effect sizes in cases with primary insomnia and insomnia comorbid to physical and mental disorders [104].

Stepped care CBT-I: The standard delivery method of CBT-I is in-person. Yet, there are alternative ways to deliver CBT-I that make the intervention more readily available and reduce costs. In 2009, Espie [105] proposed a ‘stepped care’ model for these alternative treatment modalities. The first step comprises the least intensive therapy and consists of self-administered CBT-I with minimal contact (e.g., telephone, brief appointments) or entirely self-directed CBT-I (e.g., with books, audiovisual, internet support). Lancee and colleagues [106] found moderate to large effects of unguided internet-delivered and email-delivered CBT-I on various sleep measures. Nowadays, there are various commercially available CBT-I apps on the market. Advantages of these digital platforms are: easier access, cost efficiency, provide instant support, and quick and easy insight into sleep data and progress [107]. Van der Zweerde and colleagues [108] showed that nurse-guided internet-delivered CBT-I in patients in general practice exerts large and persistent sleep improvements. When a patient shows an incomplete treatment response, he/she should be referred to the second level of ‘stepped care’, which entails nurse-delivered in-person CBT-I to small groups of patients. Higher levels of the model consist, among others, of individualized tailored CBT-I administered by a clinical psychologist or by an expert in sleep medicine, for example, from a sleep clinic.

TranS-C: Another approach is the transdiagnostic sleep and circadian (TranS-C) intervention [109]. TranS-C combines elements of CBT-I with chronotherapy and interpersonal and social rhythm therapy (ISRT). Its core modules are regularization of the sleep–wake rhythm, adjusting maladaptive sleep beliefs, improving daytime functioning, and preventing relapse/treatment consolidation. Optional modules include sleep restriction, stimulus control, targeting environmental factors complicating sleep, and treating sleep-related worry/vigilance. TranS-C also includes modules focusing on the treatment of comorbid sleep disorders such as adherence to continuous positive airway pressure therapy in cases of OSAS, adjusting delayed or advanced sleep–wake rhythms, and treating nightmares. TranS-C has been shown to increase actigraphy-measured total sleep time in adolescents with evening circadian preference [110]. Another study in patients with diverse severe mental disorders and sleep disturbances revealed that, compared with the control, TranS-C produced significant and long-lasting improvements in objective and subjective sleep outcomes [111]. 

## 4. Evaluation of Adapted Versions of CBT-I in Specific Mental Health Populations and the Elderly

### 4.1. Major Depressive Disorder

Major depressive disorder (MDD) is characterized by depressed mood, loss of pleasure, and additional symptoms such as clinically significant distress/dysfunction [72]. It is one of the most frequently occurring mental health disorders, with a lifetime prevalence of 16% in high-income countries [112]. Insomnia symptoms are experienced by the vast majority of adults with MDD [113] and about 40% suffer from insomnia disorder [114]. Insomnia is not merely a symptom or consequence of depression. It is well established that insomnia exacerbates depression severity (e.g., [115]), hinders the response to antidepressant therapy [116,117], often resides after successful depression treatment (e.g., [86,118]), and persistent insomnia poses a risk for relapse [118,119]. Considering its impact on psychopathology, insomnia comorbid to MDD needs to be treated with an effective intervention targeting disturbed sleep. 

Manber et al. [19] conducted a pilot randomized controlled trial (RCT) in participants with MDD and insomnia. All received an antidepressant and half of the participants were concomitantly treated with individual CBT-I and the other half with a control treatment. Compared with the control group, the CBT-I group exhibited large improvements in nearly all subjective and objective sleep measures and insomnia remitted in a large proportion (50% vs. 7%). A larger RCT in a comparable population confirmed that CBT-I added to pharmacotherapy for depression produces a greater reduction in insomnia severity and a higher insomnia remission rate than the control condition [20]. Group CBT-I has also been shown to be effective [21]. 

To increase access to care, various forms of self-help CBT have been developed, including therapist-guided internet-delivered CBT (ICBT). Blom and colleagues [22] treated patients with depression and insomnia with ICBT for depression (ICBT-D) or for insomnia (ICBT-I). While both interventions equally ameliorated depression symptoms, ICBT-I more favorably improved insomnia severity (remission rates 57% vs. 19%) and diary-based sleep latency and efficiency. Another guided online CBT-I, i-Sleep, was delivered to persons with depressive symptoms and insomnia. Compared with the control, i-Sleep significantly and persistently decreased insomnia severity (treatment response: 64% vs. 9%, remission: 62% vs. 15%), subjective sleep latency as well as wakefulness after sleep onset, and increased sleep efficiency [17]. To increase practicality, CBT-I has been condensed to a brief intervention (BCBT-I; ca. four sessions). Carney et al. [23] investigated the effectiveness of individual BCBT-I in patients with MDD and insomnia. They received BCBT-I and an antidepressant or placebo, or sleep hygiene and an antidepressant. The three groups reported comparable large sleep improvements; yet, PSG-based total wake time decreased in the BCBT-I groups and increased in the sleep hygiene group. Thus, subjective measurements may obscure the sleep disruptive effects of various antidepressants. An RCT study found that group BCBT-I effectively reduced insomnia symptoms in people with depression and insomnia [24]. An even briefer format of BCBT-I (two short in-person sessions and two very short phone calls) was examined in veterans with co-occurring depression and insomnia [25]. Though sleep improved after both BCBT- I and control treatment, the first evoked larger improvements in insomnia severity (treatment response: 55% vs. 21%) and diary-based number of awakenings and sleep efficiency. In a sample of individuals with residual depression and persistent insomnia, individual BBT-I produced higher remission rates for insomnia (50% vs. 0%) and depression (50% vs. 6%) than treatment as usual [26].

A youth version of BCBT-I (three to four sessions) has been developed, with age-appropriate modifications such as a greater focus on limiting bedtime use of electronics and on the adverse impact of poor sleep on youth-appealing outcomes, such as peer relationships. In a pilot RCT, adolescents with depression and insomnia received a combination of CBT-D and youth-adapted BCBT-I or sleep hygiene [27]. Co-treatment with BCBT-I induced medium to large improvements in objective and subjective sleep variables that did not differ significantly from the sleep hygiene control condition. For instance, insomnia remission rates were 60% and 43%, respectively. A group CBT-I modified to meet the needs of adolescents was found to moderately decrease insomnia severity in a small group of depressed adolescents, but the study did not involve a control group [28]. 

Goldschmied and Gehrman [120] recently described several adaptations of CBT-I based on clinical experience to specifically address depression symptomatology, thereby possibly enhancing its effects on both insomnia and depression severity. Although not evidence-based, the suggested modifications are appealing with a number of illustrations of adaptations: persons with MDD are often not motivated to participate in activities, incorporating cognitive–behavioral approaches targeting low motivation may increase adherence to behavioral treatment recommendations; depressogenic thoughts or beliefs that interfere with the treatment, such as ‘There is no point in engaging in CBT-I’, should be tackled with cognitive restructuring. In case of diurnal mood variations, it may help to schedule pleasurable activities and/or light therapy at the time of day when one feels worse. 

Concluding, CBT-I, both in the individual and the group format, improves sleep in depressive persons with a large effect size. Internet-guided as well as shorter (but not too short) versions of CBT-I may produce comparable effects. Though results need to be corroborated, evaluation of BBT-I in patients with MDD is also promising. Concerning CBT-I modifications, the evaluation of the CBT-I versions adapted to adolescents with MDD awaits studies with larger sample sizes and appropriate control groups. Finally, depending on their relevance for individual MDD patients with insomnia, the adaptations described above can easily be implemented into standard CBT-I.

### 4.2. Bipolar Disorders

A bipolar disorder (BD) is a mood disorder characterized by unusual mood shifts in which periods of depressive mood with lethargy alternate with periods of mania with typical abnormally elevated mood and energy levels [72]. Bipolar type 1 disorder is characterized by one or more manic episodes and possibly depressive episodes. Bipolar type 2 disorder is characterized by at least one major depressive and one hypomanic episode. Sleep disturbances are core symptoms of BD [121] and changes in sleep patterns are incorporated in the DSM-5 diagnostic criteria [72]. It is estimated that more than 40% of persons with BD suffer from insomnia disorder [122], which underlines the need for access to insomnia treatment in this population. 

Kaplan and Harvey [29] investigated the tolerability and safety of behavioral treatment of insomnia adapted to BD type 1 patients with insomnia. The eight-session therapy consists of sleep education, stimulus control, winding-down and wake-up routines around a set bedtime and rise time (sleep–wake regularization), and, when necessary, sleep restriction. Although the study is under-sampled and lacks a control group, the insomnia therapy clearly improved sleep. Harvey and colleagues [30] developed a modified version of CBT-I for patients with BD type 1 (CBT-I-BD). One modification in sleep restriction was that the limitation of time in bed was replaced by regularization of the sleep–wake rhythm. Furthermore, a minimum of 6.5 h in bed was maintained to avoid triggering (hypo)mania. Within stimulus control, the emphasis on going to bed when feeling sleepy was skipped, since persons with BD often do not experience sleepiness when they shift to a (hypo)manic episode. Additionally, elements of ISRT (activity scheduling), chronotherapy (exposure to dim light before bedtime, bright light directly after waking up, sleep compression), and motivational interviewing were incorporated. CBT-I-BD was tested against psychoeducation-only in persons with BD type 1 who were inter-episodic at the time of treatment. Compared with the control, CBT-I-BD had stronger effects on insomnia severity (ISI) (treatment response: 68% vs. 29% and remission: 73% vs. 14%). There were no significant differences in attrition between the two groups (11% dropped out during treatment). In a study by Kaplan and colleagues [123] that built on the latter study [30], a subset of patients received the RISE-UP routine alongside CBT-I-BD in session one. The aim of this routine was to reduce sleep inertia in the morning, which seems to occur in 42% of patients with BD, by refraining from snoozing, exposure to sunlight, planning social appointments, cold showers, upbeat music, and seeking activity in the first hour after awakening. Significantly higher activity levels in the morning and shorter duration and milder sleep inertia were found in the CBT-I-BD plus RISE-UP group relative to the control group.

Taken together, there is a lack of high-powered RCTs on CBT-I in persons with BD. The only published RCT showed efficacy of CBT-I adapted to BD patients. In this study, it became apparent that features of other sleep disorders (hypersomnia, delayed sleep phase disorder) were commonly observed alongside insomnia, which warrants revisions in how sleep-diary measures are interpreted as outcome measures. Adaptations within traditional CBT-I components were particularly aimed at avoiding mania and sleep inertia. As no study directly compared standard CBT-I with CBT-I-BD, the added value of the bipolar-specific modifications is unknown. 

### 4.3. Anxiety Disorders

Anxiety disorders are characterized by excessive fear and anxiety and related behavioral disturbances. This group of disorders includes generalized anxiety disorder, specific phobia, social phobia (social anxiety disorder), agoraphobia, panic disorder, and separation anxiety disorder [72]. The lifetime prevalence of any anxiety disorder is 29%, with prevalence rates for the specific types of anxiety disorders varying between 2–12% [124]. Sleep problems, in particular insomnia, are often experienced by individuals with an anxiety disorder: the prevalence of life-time insomnia disorder comorbid to an anxiety disorder is around 30% [125]. Both experimental and naturalistic studies suggest that insomnia enhances anxiety [126,127]; thus, treatment of insomnia in individuals with an anxiety disorder is paramount. 

In one of the few studies assessing the effects of CBT-I in persons with an anxiety disorder, a direct comparison was made between CBT-I and CBT for generalized anxiety disorder (CBT-GAD) [31]. All participants received both treatments, but the order in which the therapies were presented was randomized. The group sizes were small and treatment duration was significantly longer than usual (up to 30 sessions), which seriously limits the conclusions of this study. Nevertheless, CBT-I was shown to improve all subjective sleep outcomes with large effect size, whereas CBT-GAD solely improved subjective sleep quality (PSQI). One third of the patients who received CBT-I first reached insomnia recovery, whereas none of the CBT-GAD-treated patients met that criterion. Although somewhat superior for sleep symptoms, the CBT-I treatment was clearly inferior in treating anxiety symptoms. With a dropout rate close to 30% in the CBT-I group, the attrition was not optimal, but this might be related to the high number of treatment sessions. 

Additional evidence for the effectivity of standard CBT-I in persons with an anxiety disorder comes from a study from Bélanger and colleagues [32]. They showed no differences between individuals with only a chronic insomnia diagnosis and those with a comorbid anxiety or depression diagnosis in CBT-I-induced improvements in ISI scores (treatment response: 71% vs. 81%) or in the dropout rates (6% in both). Unfortunately, the study did not differentiate between the types of comorbid disorders, but, of those with a comorbidity, the majority had an anxiety disorder diagnosis (76%). 

To our knowledge there are no scientifically assessed CBT-I protocols that were specifically adapted to individuals with a comorbid anxiety disorder. However, an alternative psychotherapeutic approach used in persons with anxiety disorders is a mindfulness-based cognitive therapy for insomnia. Although no clinical insomnia symptoms (baseline PSQI range 2–10) were required for participation in their study, Yook et al. [33] showed improved sleep quality with a large effect size with this therapy.

Overall, evidence is limited, but the standard CBT-I seems efficient in treating insomnia in persons with an anxiety disorder. The seeming lack of CBT-I protocols specifically adapted to anxiety disorders may suggest that there is no need for an adapted protocol for this population. However, since most studies paying attention to comorbid insomnia and anxiety disorder focus on depression as well and do not differentiate between those two populations, studies focusing on anxiety disorder specifically may be needed to better evaluate whether an adaptation to the standard CBT-I is beneficial. 

### 4.4. Posttraumatic Stress Disorder

Posttraumatic stress disorder (PTSD) is triggered by a terrifying event, producing persistent symptoms such as flashbacks, nightmares, severe anxiety, and uncontrollable thoughts about the trauma that interfere with daytime functioning [72]. Combined population studies found an average lifetime PTSD prevalence of 3.2% in men and 7.8% in women [128]. The vast majority of PTSD patients experience sleep disturbances, most frequently insomnia: approximately 70% report insomnia symptoms and 40% is affected by insomnia disorder [129]. Insomnia is an established risk factor for the development of and relapse in PTSD, while interventions targeting insomnia not only improve sleep quality but also ameliorate daytime PTSD symptoms [130]. 

An RCT in a community sample of individuals with PTSD and insomnia demonstrated that, compared with a control group, individually provided standard CBT-I produced large improvements particularly in subjective sleep variables, such as sleep latency, sleep efficiency, total sleep time, sleep quality, and insomnia severity [34]. Insomnia remitted in 41% of the CBT-I and in 0% of the control group. A small RCT in veterans with PTSD, insomnia, and nightmares found that individual CBT-I combined with imagery rehearsal therapy (IRT) for nightmares resulted in medium to large improvements in insomnia severity (ISI), sleep quality (PSQI), and all diary-based outcome measures [35]. Insomnia remitted in 33% of the intervention and in none of the control group. A total of 33% dropped out of the intervention. Another study with veterans with PTSD and insomnia investigated the effects of group-format CBT-I delivered either in person or, to increase access to care, via video telehealth (camera in group room) [36]. Both conditions were associated with high attrition rates typical for this complicated population (54% completed in-person and 47% telehealth treatment) and modest improvements in insomnia severity and sleep quality. BBT-I-MV is a BBT-I version tailored to military veterans, in which language is culturally adapted to military samples, sleep education is limited to principles underlying stimulus control and sleep restriction, and these principles are illustrated by military-relevant examples [37]. An RCT in combat-exposed veterans with insomnia disorder and cases of PTSD revealed that the large majority (85%) completed all BBT-I-MV sessions. Further, compared with the control condition, BBT-I-MV produced slightly higher rates of response (77% vs. 50%) and remission (53% vs. 31%) and exerted significantly greater effects on insomnia severity and sleep quality. A direct comparison of BBT-I adapted to veterans (unclear whether this is similar to BBT-I-MV) and CBT-I delivered to veterans with insomnia disorder found dropout rates of 29% and 50%, respectively, and revealed large improvements in insomnia severity, subjective sleep quality, diary-derived sleep latency, wake after sleep onset, and sleep efficiency after both treatments [38].

In conclusion, though the findings need to be corroborated in larger samples, CBT-I appears an efficacious and feasible therapy for insomnia comorbid to PTSD. However, in veterans with PTSD and insomnia, CBT-I attrition rates are high, while it may produce relatively moderate sleep improvements. BBT-I with fewer and shorter sessions and tailored to this particular group of PTSD patients seems promising, both with regard to treatment adherence and to treatment response.

### 4.5. Substance Use Disorders

Substance use disorders (SUD) are a group of disorders characterized by the persistent use of drugs or alcohol despite substantial harm and adverse consequences [72]. This diagnosis is relatively common, with, for example, a prevalence of 7.4% in the United States [131]. Sleep disorders, and specifically insomnia disorder, are often found in SUD patients. Prevalence rates of insomnia symptoms (as defined by [72]) range between 47% to 78% depending on the type of substance used, with the highest prevalence in benzodiazepine dependency and the lowest in cannabis dependency [132]. 

The majority of studies on the effects of CBT-I focus on populations with an alcohol-use disorder. In addition to standard CBT-I, an adapted protocol including a module with psychoeducation and cognitive elements focused on the effects of alcohol use and withdrawal on sleep is available. In a small RCT, this protocol was shown to effectively reduce ISI scores (83% of treatment completers reached remission of insomnia) and diary-based sleep latency and increase sleep efficiency (all large effect sizes), without affecting subjective sleep duration [39]. A subsequent study showed similar results for BCBT-I aimed at adolescents [40]. In recovered alcoholics a comparable BCBT-I protocol resulted in improved subjective sleep efficiency, sleep latency, and sleep quality, with remission achieved in 75% when using sleep latency (<30 min) and 38% using the sleep quality criteria (PSQI < 6) [41]. Dropout rates in both standard CBT-I and the alcohol use adapted CBT-I treatments did not differ from the control condition ([41]: 20%, [39]: 27%, [40]: 36%). 

The effects of CBT-I were also assessed in patients with a hypnotics’ dependency. Standard CBT-I was shown to improve objective and subjective sleep latency but no other sleep measures [42]. Taylor and colleagues [43] assessed the effects of a group CBT-I protocol with an added medication reduction module. Significant reductions in ISI scores in the CBT-I group compared with TAU were observed. Insomnia remission (ISI ≤ 14) was reached in 54% of the CBT-I treated participants (0% in TAU), yet there were no differences in hypnotics’ use. The dropout rate was equally low, around 10% in both groups. 

Overall, CBT-I, standard or adapted, improves sleep in persons with SUD. Although no direct comparisons between standard CBT-I- and CBT-I SUD-adapted protocols are available, and thus the added value of the adaptations cannot be ascertained, it is likely that modifications that focus on specific consequences of substance use and withdrawal on sleep are beneficial in the treatment of this population. In future studies, one might evaluate whether the SUD-tailored protocol helps the patient to have more realistic ideas of what to expect from their sleep during the different stages of their substance-use healing process.

### 4.6. Schizophrenia Spectrum Disorders

Schizophrenia spectrum disorders (SSD) are characterized by positive symptoms such as delusions, hallucinations, disorganized and/or abnormal thinking and/or behavior, and negative symptoms such as apathy, anhedonia, and blunted affect [72]. It has a lifetime prevalence of 0.75% [133]. With up to 80% of persons with SSD reporting insomnia symptoms [134] and prevalence of insomnia disorder ranging from 19% to 44% [135,136], insomnia is common. Sleep disturbances often start in a prodromal phase of SSD and are associated with the development and exacerbation of psychotic symptoms [137]. 

Hwang and colleagues [44] tested a group-based BCBT-I in a non-randomized trial in schizophrenia patients. Compared with TAU, the BCBT-I significantly improved insomnia severity (medium effect size), total sleep time, sleep latency, sleep efficiency, sleep quality, and daytime dysfunction (all small effect sizes). There were no dropouts during the study. 

Freeman and colleagues [45] evaluated an eight session CBT-I adapted to SSD in patients who experienced delusions or hallucinations at the time. Standard CBT-I components were complemented with components that addressed, among others, delusions and hallucinations that impair sleep, excessive sleeping as a strategy to escape voices, fears related to negative experiences with the bed, and circadian disruptions (see [138] for a comprehensive overview). The authors did not test for statistical significance but reported descriptive statistics and effect sizes instead. Relative to TAU controls, persons treated with adapted CBT-I had larger improvements in insomnia severity (large effect size), sleep latency (medium effect size), wakefulness after sleep onset and total sleep time (small effect size). Furthermore, a larger proportion of the CBT-I treated reached insomnia remission (41% vs. 4%).

In conclusion, both BCBT-I and individual CBT-I tailored to SSD seem effective in treating insomnia in patients with SSD. For now, more (adequately powered) studies are necessary to appropriately appraise the efficacy of (adapted) CBT-I in this population. For future studies we may bear in mind that differences in responsivity to CBT-I exist between clusters of SSD patients, based on factors such as baseline insomnia symptoms, presence of other sleep disorders [139], psychopathology, and antipsychotic use [140]. 

### 4.7. Attention Deficit Hyperactivity Disorder

ADHD is characterized by a persistent pattern of inattention and/or hyperactivity/impulsivity that interferes with functioning or development [72]. According to a large meta-analysis the prevalence of ADHD in children is 7% [141]. For 60% of these children, ADHD persists into adulthood [142], where occurrence is estimated at 5% [143]. Insomnia disorder is very common is this patient group [144]. Furthermore, several studies report associations between ADHD and insomnia symptoms, yet the direction and strength of these associations remain to be tested (reviewed in [145]). 

In a small feasibility study, Jernelöv and colleagues [46] investigated whether insomnia symptoms improved following a novel group (6–10 patients/group) CBT-I intervention adjusted to patients with ADHD (CBT-I-ADHD). As delayed sleep phase disorder is common in ADHD [146], adjustments consisted of addition of behavioral components targeting delayed sleep phase, insufficient sleep hygiene, and problems with waking up in the morning. Furthermore, components targeting commonly occurring difficulties with planning and organizing were added. Finally, adjustments in format were made such as shortening of session length, prolongation of the treatment duration (10 weeks), and inclusion of a short telephone call between sessions. This protocol was shown to significantly improve ISI scores with a large effect size that remained at the 3-month follow-up. Insomnia remission was reached by 32% posttreatment and 42% at follow-up. With a dropout rate of 11%, therapy adherence was good. 

Various of the adaptations of CBT-I-ADHD are similar to components of the TranS-C approach [147], making TranS-C itself a promising therapy for ADHD populations. In a recent pilot open trial of TranS-C in adolescents with ADHD, six individual sessions, with four core, four ‘cross-cutting’, and seven optional modules, were given [47]. PSQI scores decreased significantly at posttreatment and subjective sleep latency improved with a large effect size; these effects persisted through follow-up. However, objectively measured time in bed increased significantly.

Overall, CBT-I-ADHD and TranS-C seem promising approaches to treat insomnia in people with ADHD. However, the results of the studies described above should be interpreted with caution because no control groups were used, and the sample sizes were small. Larger RCTs are necessary to test the efficacy of these therapies. 

### 4.8. Autism Spectrum Disorder

Autism spectrum disorder (ASD) is characterized by persistent deficits in social communication and interaction across multiple contexts. Specifically, deficits are seen in social–emotional reciprocity, nonverbal communicative behaviors used for social interaction, and in developing, maintaining, and understanding relationships [72]. Symptoms of ASD present early in life and the disorder is usually diagnosed in childhood. Prevalence rates of ASD vary largely by geographic area and time of assessment and estimates range between 0.02–1.5% [148]. 

To our knowledge, there are no studies aimed at CBT-I in adults with ASD, therefore this section focuses on insomnia treatment in children with ASD. Although estimates of the prevalence of sleep problems in children with ASD differ largely (50–80%), disturbed sleep seems prevalent, with trouble falling asleep being the most common complaint [149,150]. Treatment of insomnia in ASD children generally focuses on both the parents and the children and an adapted CBT-I protocol for children (CBT for childhood insomnia, CBT-CI) has been developed and proven effective [151]. This six-session protocol contains the following elements: sleep education, sleep hygiene, bedtime fading techniques (slowly moving the child’s bedtime to desired time), challenging maladaptive thoughts, graduate exposure therapy for fear around sleep and separation from parents, and relaxation techniques. An RCT tested the effects of CBT-CI or melatonin treatment in a population of 4–10-year-old children with ASD. CBT-CI improved objective (actigraphy) and subjective sleep quality with medium effect sizes. Although CBT-CI seemed effective, the percentage of children reaching insomnia remission was limited to about 10% [48]. McCrea and colleagues [49] adapted the CBT-CI protocol to better fit the cognitive level and specific symptoms of ASD by adding visual aids to the verbal sleep education, using antecedent strategies (preventing behaviors from occurring by modification of the environment), having specific attention for sensory problems (hyper- or hyposensitivity to sights, sounds, smells, tastes, or textures), and employing simplified cognitive and relaxation tasks. In a longitudinal study in 6–12-year-old children with ASD, this protocol showed improvements with large effect sizes in subjective and objective measurements of sleep latency, total sleep time, and sleep efficiency. Furthermore, 75% of treatment non-completers (12% dropout) and 92% of treatment completers did remit from insomnia. Although the treatment was well received, there were some concerns about the usability due to the large commitment that is asked from patients, parents, and psychologists. Therefore, a briefer protocol has been developed that can be administered either face-to-face or remotely [152], but this protocol remains to be tested. 

In sum, for the primary school age group there are promising ASD adapted CBT-I protocols available that might only require some additional tweaking to improve usability and accessibly. Although several symptoms of ASD remain into adulthood, to our knowledge similar protocols for adult ASD patients with insomnia have not been tested. In all likelihood, when treating insomnia, paying specific attention to, for example, the sensory problems associated with ASD may also prove beneficial for adult ASD patients.

### 4.9. CBT-I in Older People

The population of the elderly grows steadily. According to the United Nations, aging is a global phenomenon, and the population aged 65 or older is projected to double from 703 million in 2019 to 1.5 billion in 2050 (United Nations, World Population Aging, 2019). Insomnia occurs more often in older age: the prevalence of insomnia symptoms in the elderly ranges from 30% to 48% [153] and of insomnia disorder from 12% to 20% [154]. Over the years, evidence has accumulated that CBT-I is an effective and safe first-line treatment in healthy elderly with insomnia [155,156]. An RCT in older adults with primary insomnia revealed that individually provided standard CBT-I significantly improved objective (PSG) total wake time, sleep efficiency, and amount of N3 compared with both Zopiclone and the placebo control groups [50]. A recent longitudinal study [51] demonstrated that also in the highest age groups, over 75 years old, CBT-I is a very effective treatment. The largest effects were observed in insomnia severity, followed by subjective sleep efficiency, sleep latency, and wakefulness after sleep onset. 

Because the elderly population is a heterogeneous group more likely to have functional or sensory impairments and medical, cognitive, and/or psychiatric comorbidities, it is important to consider different modifications and delivery methods to enhance the ability to engage successfully and safely in CBT-I. An alternative approach for stimulus control is called counter control [157]. Because older adults cannot always get up by themselves, it is advised to stay in bed during the night when awake and engage instead in a relaxing activity that is not related to falling asleep [158]. Furthermore, additional effort is needed to limit daytime sleeping outside the bedroom. Sometimes it is necessary to avoid the situation where they habitually fall asleep. It can be difficult for older adults to restrict sleep all at once; instead of abrupt sleep restriction, sleep compression is used to gradually decrease the time in bed. Herein, the amount of sleep is initially restricted based on the time spent in bed rather than the traditionally used total sleep time [159]. Of note, especially in this population, it is important to consider time spent sleeping during the day when calculating the advised time in bed. 

The efficacy of BBT-I was first studied in older adults by Buysse and colleagues [52]. Compared with information control, BBT-I decreased both subjective and objective sleep latency and wakefulness after sleep onset and increased sleep efficiency, as well as subjective sleep quality with moderate effect sizes. The therapy seemed well received with a dropout rate of only 8% and 55% reached insomnia remission. McCrae and colleagues [53] also tested the effects of an in-person delivered BBT-I, but, in contrast to previous studies, the time spent with a therapist was matched between the treatment and the control groups. Unlike the self-monitoring control, BBT-I improved subjective sleep latency, wakefulness after sleep onset, sleep efficiency, and sleep quality (all medium to large effect sizes) and these improvements were maintained at follow-up.

Bothersome pain is a common complaint in older adults, reported by about 53% [160], and a likely risk factor for insomnia [161]. Osteoarthritis, a disease accompanied by pain, affects almost 50% of the elderly population [162]. Strikingly, more than half of the osteoarthritis-affected elderly experience sleep problems [163]. Recently, a large RCT tested a brief telephone CBT-I in older adults with moderate to severe insomnia and comorbid osteoarthritic pain [54]. The intervention consisted of six 20–30 min telephone sessions provided over eight weeks. After 2 months, ISI scores improved significantly in the telephone CBT-I group compared with the educational control group. Twelve months later, 56% of the CBT-I group remained in remission compared with 29% of the educational control group.

Sadler and colleagues [55] explored whether a hybrid form of CBT-I (advanced CBT-I) for older adults with comorbid depression produced better outcomes. The advanced CBT-I included three additional strategies targeting comorbid depression: behavioral activation, cognitive reframing for depression, and positive affirmations to increase hopefulness. Compared with the control group, both standard and advanced CBT-I resulted in significantly larger improvements in all subjective sleep variables, insomnia severity, and depression symptomatology. Both types of CBT-I did not differ from each other and thus seemed to be equally effective at reducing comorbid insomnia in older adults with depression.

A decline in cognitive abilities is another factor to take into account when treating older adults with insomnia. Mild cognitive impairment (MCI) is a transitional stage between expected cognitive decline of normal aging and dementia [164]. It impacts almost 20% of older adults, and sleep problems are relatively common in this group with prevalence rates from 14% to 59% [165]. Cassidy-Eagle and colleagues [56] performed a pilot study in which a six-session adapted version of CBT-I was administered to older adults with MCI. Adjustments included a decrease of the cognitive components, handouts printed with larger font size for visually impaired participants, and larger writing spaces for patients with deficits in fine motor skills. Learning and memory aids were used to enhance understanding of the content. Insomnia severity and actigraphy-based sleep latency, wake after sleep onset, and sleep efficiency improved significantly with large effect sizes compared with the control group.

Overall, there is growing evidence that older adults benefit from CBT-I and BBT-I. Adaptations in the standard protocols are useful to meet the needs of elderly with differing comorbid conditions. Over the years, various adaptations of CBT-I have been developed for older adults that seem beneficial. Although the efficacy of internet-based programs is rarely studied in older adults, it can be necessary to expand access to this sleep therapy in this group. Implementing CBT-I/BBT-I into different settings (residential care, different community/home settings) and possibly being administered by nursing staff would increase access to this effective treatment for the elderly.

## 5. Overarching Conclusions

Consistent with previous reports [16,84,89,166], our overview indicates that CBT-I is efficient in reducing insomnia symptoms across different mental health populations. Large treatment responses and high remission rates are reported in these complex patients. For improvement of adherence, better access, or financial avail, a brief version of CBT-I (BCBT-I or BBTI) is often chosen in mental health settings. Typically, these brief protocols exert similar, albeit sometimes more modest, positive effects on insomnia symptoms, treatment response, and recovery rate. As a direct comparison between the efficacy of CBT-I and briefer versions within a study is often not available, it is difficult to draw hard conclusions on its benefit beyond the standard protocol. However, comparing between studies, it seems that BCBT-I may benefit adherence in some patient groups (schizophrenia and veterans with PTSD), while in other groups this was not obvious (SUD, depression) or even may have poorer results (anxiety). Focusing on accessibility, when, for example, only one therapist is available for both the sleep and psychopathology treatment, the use of BCBT-I (or BBT-I) might be more efficient, making combined treatment feasible. Especially, BBT-I is more applicable in settings such as primary health care offices or (psychiatric) nursing wards as it can be delivered by health care professionals without a psychotherapy degree (e.g., psychiatric nurses and physician assistants).

In our opinion, the most reported adaptations to CBT-I are highly relevant and applicable. The availability of these adaptations offers the opportunity to personalize the treatment protocol to the individual needs of the patient based on pathology specific recommendations. Hertenstein and colleagues [167] argued that the heterogeneity between individuals is so large—even within specific diagnostic groups—that disorder specific adaptation may prove redundant. In contrast to this notion, we pose that these adapted protocols may provide elegant and readily available tools to handle commonly observed problems in the applicable comorbid condition. By using a carefully designed and evaluated adapted protocol, therapists do not have to reinvent the wheel when designing their treatment plan for individual patients; they may just need to slightly tweak the protocol to make it an exact fit for their patient. 

### 5.1. Gaps in the Current Literature

It appears that sufficient knowledge on CBT-I (adaptations) is available for the depression and elderly populations to provide suitable treatment options for these patient groups. Nonetheless, for all mental disorders associated with insomnia, with the possible exception of depression, the number of effectivity studies of an adapted protocol was limited to two, one, or even zero. Furthermore, the available studies were often of a small sample size and/or without an (active) control group. Moreover, within-group heterogeneity was often not taken into account in these studies. For schizophrenia spectrum and bipolar disorder, no differentiation was made between the different subpopulations within the diagnosis group, whereas for others, available studies are limited to or mainly focused on a specific subpopulation. Within SUD, only protocols aimed at alcoholism and hypnotics’ use are available, while adaptations for other substances—cannabis, opioids, or stimulants—are not reported. Within PTSD, the main focus lies on PTSD in (war) veterans, yet the effects of CBT-I (and its adaptations) might be different in PSTD patients with other traumatic experiences (e.g., early life trauma). As insomnia is equally prevalent and detrimental in those patients, this is a serious omission in knowledge. Similarly, ASD protocols are limited to children with ASD, whereas evidence for benefits of CBT-I in adults with ASD is lacking. Since adults with ASD, similar to those with ADHD, often suffer from insomnia plus delayed sleep phase disorder, it might be worthwhile to specifically investigate TranS-C in this group. Notably, there is a possibility that there are successfully adapted protocols available for clinical use that are not published in scientific journals.

### 5.2. Perspective

The gaps in the literature mentioned above highlight possibilities for future studies and novel treatment strategies. Within the studies covered in this review, it became apparent that a large part of published articles concerned pilot RCTs that were not followed by an adequately sized study, even if this was intended. This is unfortunate, as pilot studies often do not provide accurately estimated population effect sizes and, therefore, are at a substantial risk of containing false positive or false negative results [168]. It mostly remains unclear why the large RCT is lacking. We propose two potential reasons. It could be that larger studies that do not replicate the effects of earlier published (positive) pilot studies are not offered for publication, resulting in confirmation bias due to selective availability of mostly positive treatment results. In a recent review on CBT-I applied to those with psychiatric comorbidity, Hertenstein and colleagues [167] found indications for such a publication bias and argued that the small sample sizes in available studies might have inflated effect sizes. In addition, as the pattern of small-sampled studies being more likely to report positive effects extends to meta-analytical estimation [169], an even stronger overestimation of the effects of CBT-I may be expected in meta-analyses [170]. Another explanation for the lack of large-scale RCTs is that these are not always feasible in complex mental health populations. Commonly heard barriers in mental health patients are reluctance in simultaneous treatment of two disorders and a general hesitance in participating in scientific research. Reaching the desired sample size may rely on multicenter studies, which are laborious, resource intensive, and therefore not achievable for most researchers. As a result, adapted CBT-I protocols may be constructed time and again by individual therapists, while comparably tailored protocols are possibly already being used but not published. Due to the under-sampled studies, robust conclusions on the efficacy of CBT-I and the added value of its adaptations in specific mental health populations remain out of reach. Here, we propose an approach for breaking the current impasse, namely the construction of an international database containing *n* = 1 studies. This database would consist of (adapted) CBT-I protocols for specific populations and detailed information on demographic, psychiatric, and sleep (insomnia) characteristics and relevant outcome measures. When a clinician would want to utilize adaptations for a specific patient, one could sign up to access the respective protocol. Within the database, existing reporting standards for *n* = 1 studies, such as those listed by Vohra and colleagues [171], should be maintained. The aggregation of *n* = 1 trials can provide population estimates of group effects [172], which over time enables researchers to assess treatment response and remission after a certain protocol in a large group of relevant patients. Additionally, potential differences in response to CBT-I treatment based on personal, mental health, and sleep disturbance factors could be investigated using aggregated data with, for example, clustering analyses. Moreover, *n* = 1 studies included in the database could in themselves be valuable, especially when multiple comorbidities and large interindividual differences exist, as is often the case within mental health populations, because these cases are often excluded in RCTs [167,173]. Naturally, this proposal comes with its own challenges such as heterogeneity in assessed populations and languages, problems in quality control such as potential protocol misalignments between different mental health care institutes, and the burden of structural database maintenance (e.g., financial, time, and staff). In populations where disorder-specific CBT-I adaptations are lacking, expert panels could be formed in which professionals with knowledge of sleep medicine and specific psychopathology collaborate. These panels could then set out plans to adapt CBT-I protocols to better meet the therapeutic needs of specific populations.

In mental health patient groups, CBT-I produces good remission rates, certainly in comparison with pharmacotherapy in general. Yet, in the studies described in this review between 27% and 68% of those treated with a version of CBT-I did not reach insomnia remission. In view of the negative consequences of insomnia on mental wellbeing, this relatively large percentage of CBT-I resistant insomnia is of particular concern in psychiatric populations. Therefore, besides expansion of available CBT-I adaptations, it is also important to develop alternative treatment strategies (either pharmacotherapeutic, psychotherapeutic, or combinations thereof) for patients with refractory insomnia after CBT-I.

## Figures and Tables

**Table 1 jpm-12-01070-t001:** Summary of articles describing (adapted) cognitive behavioral therapy protocols for insomnia in different mental health populations and the elderly.

Author	N	Country Age % Females	Diagnosis	Treatment Group	Control Group	Sleep Measures	Follow-Up	Results
Major depressive disorder								
Manber et al., 2008 [19]	30	USA 48.6 ± 13.3 61%	MDD (DSM-IV-TR and HRSD_17_ > 14) and insomnia disorder (DSISD and sleep diary-based: SOL and/or WASO > 30 min at least 3 times/week and TST ≤ 6.5 h at least 3 times/week)	Antidepressant plus CBT-I	Antidepressant plus quasi-desensitization	Objective: actigraphy (TWT, TST, SE) Subjective: ISI; sleep diary (TWT, TST, SE, SSQ)	None	Compared with control, CBT-I produced insignificantly larger improvements in ISI and all actigraphy- and diary-based sleep variables, except for TST.
Manber et al., 2016 [20]	150	USA 46.6 ± 12.6 73.3%	MDD (DSM-IV-TR and HDRS ≥ 16) and insomnia disorder (DSISD and ISI ≥ 11)	Antidepressant plus CBT-I	Antidepressant plus quasi-desensitization	Subjective: ISI	None	CBT-I evoked significantly larger reductions in insomnia severity (ISI) than control.
Manber et al., 2011 [21]	301	USA 49.6 ± 13.9 57.5%	MDD (DSM-IV-TR) ‘low depression’ (BDI < 14) 60%, ‘high depression’ 40%, and ‘initial complaint of insomnia’	Antidepressant plus CBT-I	None	Subjective: ISI; sleep diary (SOL, WASO, TST, SE)	None	The low and high depression groups equally benefited from CBT-I on all variables. No effect on TST.
Blom et al., 2015 [22]	43	Sweden NR 53%	MDD (DSM-IV-TR) and insomnia disorder (ISI > 10 and sleep problems >3 mo)	ICBT -I	ICBT for depression (ICBT-D)	Subjective: ISI; sleep diary (SOL, TST, SE, SSQ)	6-mo 12-mo	In both groups ISI declined from pre- to post-treatment and remained low during follow-up. Reductions in ICBT-I exceeded those in ICBT-D group. SOL and SE improved during treatment, particularly in ICBT-I.
Van der Zweerde et al., 2019 [17]	104	The Netherlands 46.0 ± 12.3 82%	At least subclinical depressive symptoms (PHQ-9 > 4) and insomnia disorder (DSM-5)	ICBT-I ‘i-Sleep’	Diary monitoring only	Subjective: ISI; sleep diary (SOL, TST, SE, WASO)	3-mo 6-mo	Compared with the control condition, i-Sleep long lastingly improved insomnia severity (ISI) and diary-based SOL, SE, and WASO.
Carney et al., 2017 [23]	107	Canada 42.3 ± 11.4 68%	MDD (DSM-IV-TR and HAMD17 ≥ 15) and insomnia disorder (insomnia complaint > 1 mo, ISI ≥ 15 and sleep diary-based: TWT ≥ 60 min and SE < 85%)	Antidepressant plus BCBT-I (4 sessions) Placebo plus BCBT-I	Antidepressant plus sleep hygiene control	Objective: PSG (TWT, SE) Subjective: ISI; sleep diary (TWT, SE)	6-mo	All groups exhibited pre- to posttreatment improvements in insomnia severity (ISI) and diary-based TWT and SE. Group differences were found for PSG-based TWT: it decreased in placebo + BCBT-I but worsened in antidepressant plus sleep hygiene control.
Norell-Clarke et al., 2015 [24]	64	Sweden NR 77%	Depressive symptomatology (BDI-II > 13), 64% MDD (DSM-IV), and insomnia disorder (DSISD and ISI > 10)	BCBT-I (4 sessions)	Relaxation training control	Subjective: ISI; sleep diary (SOL, WASO, EMW, TST, SQ)	6-mo	Both groups reported pre- to posttreatment improvements on most sleep variables. BCBT-I had significantly better outcomes on ISI, SOL, and WASO.
Pigeon et al., 2017 [25]	27	USA 58.5 ± 9.6 11%	Veterans with MDD diagnosis and insomnia disorder (DSM-IV-TR and ISI ≥ 10)	BCBT-I (4 short sessions of which 2 phone meetings)	Sleep hygiene and education control	Subjective: ISI; sleep diary (SOL, NAWAKE, WASO, TST, SE)	3-mo	BCBT-I group exhibited marginally greater pre-posttreatment improvements on ISI, WASO, NAWAKE, and SE.
Watanabe et al., 2011 [26]	37	Japan 50.5 ± 11.1 62.2%	Treatment resistant MDD (DSM-IV and GRID-HAMD > 8 and <23) and insomnia symptomatology (ISI ≥ 8)	BBT-I plus TAU for depression	TAU for depression	Subjective: ISI; PSQI (global score, SE, TST, SOL, WASO); 3 sleep items of GRID-HAMD	1-mo	Combined treatment with BBT-I produced greater improvements in ISI, PSQI global score, and SE.
Clarke et al., 2015 [27]	41	USA NR 63%	Adolescents (12–20 y) with MDD diagnosis (DSM-IV) and insomnia disorder (DSISD)	CBT-D plus youth-adapted BCBT-I (3–4 sessions)	CBT-D plus sleep hygiene control	Objective: actigraphy (TST, WASO, TWT, SE) Subjective: ISI; sleep diary (SOL, TST, SE, WASO)	3.5-mo	There were no significant differences between the conditions, except for a larger pre-posttreatment increase in actigraphy-based TST in the youth-adapted BCBT-I group.
Conroy et al., 2019 [28]	16	USA 17.3 ± 1.7 75%	Adolescents with depression (T-score on CDRS-R ≥ 55) and insomnia symptoms (≥30 min wakefulness on ≥3 nights per week)	CBT-I modified to adolescents	None	Objective: Actigraphy (TST, SE, WASO) Subjective: ISI; sleep diary (TST, SE)	None	ISI scores and diary-based SOL declined from pre- to posttreatment.
Bipolar disorders								
Kaplan and Harvey, 2013 [29]	15	USA 38.1 ± 11.5 NR	BD type 1 (DSM-IV-TR) and insomnia disorder	8-session BT-I adapted to BD	None	Subjective: ISI; sleep diary (at least SE)	None	BT-I adapted to BD resulted in a significant decrease in insomnia severity and a marginal increase in SE.
Harvey et al., 2015 [30]	58	USA NR 62%	BD type 1 (DSM-IV-TR and YMRS < 12, IDS-C < 24) and insomnia disorder (DSISD)	CBT-I adapted to sleep disturbances in BD (CBT-I-BD)	Psychoeducation	Subjective: ISI; PSQI; sleep diary (SOL, WASO, TST, SE)	6-mo	CBT-I-BD resulted in a significantly larger proportion of treatment responders (persistently) and insomnia remission (ISI persistently, DSISD not persistently) than psychoeducation. Reduction of total ISI score was greater after CBT-I-BD (not persistent). Both groups persistently improved on sleep quality (PSQI) and diary-based SE and TWT.
Kaplan et al., 2018 [30] Sub-analysis of Harvey et al., 2015 [30]	40	USA NR 62%	BD type 1(DSM-IV-TR and YMRS < 12, IDS-C < 24) and insomnia disorder (DSISD)	CBT-I-BD plus RISE-UP during the first treatment session	Psychoeducation	Objective: Actigraphy Subjective: ISI; SSS (sleep inertia severity); sleep diary (duration of sleep inertia)	None	Higher actigraphy-based activity levels during the first hour after awakening and a larger reduction in inertia duration and severity in the RISE UP group compared with the control group.
Anxiety disorders								
Belleville et al., 2016 [31]	12	Canada 44.5 ± 10.09 100%	GAD diagnosis (DSM-IV) and insomnia disorder (DSM-IV)	CBT-I (16 sessions) followed by CBT-GAD	CBT-GAD followed by CBT-I	Subjective: ISI; PSQI; DBAS; sleep diary (SOL, TST, WASO, SE)	3-mo	CBT-I persistently improved outcomes of all sleep questionnaires. The control treatment improved PSQI immediately after treatment, but not ISI score.
Bélanger et al., 2016 [32]	188	Canada 47.4 ± 12.6 62.2%	24% with diagnosis of an anxiety disorder or MDD (DSM-IV) and all with insomnia disorder (DSM-IV)	CBT-I	CT-I BT-I	Subjective: ISI; sleep diary (results not shown)	6-mo	The study reveals that having a comorbid DSM-IV diagnosis did not alter the positive effect of CBT-I on insomnia severity, but it significantly reduced the impact of CT-I and BT-I.
Yook et al., 2008 [33]	22	Korea 41.1 ± 6.3 42%	GAD or panic disorder diagnoses (DSM-IV) and no diagnostic criteria insomnia.	Mindfulness-based CT-I (MBCT-I)	None	Subjective: PSQI	None	The study shows improvement of the global PSQI score.
Posttraumatic stress disorder								
Talbot et al., 2014 [34]	45	USA 37.2 ± 10.5 69%	Participants from community sample in treatment for PTSD (DSM-IV) and persistent insomnia (DSISD)	Individual CBT-I TAU for PTSD	Waitlist plus TAU for PTSD	Objective: PSG (WASO, TST, SM); actigraphy (WASO, TST, SM) Subjective: ISI; PSQI; ESS; sleep diary (SOL, WASO, TST, SE)	6-mo	Compared with waitlist control, CBT-I persistently improved insomnia severity (ISI), subjective sleep quality (PSQI), daytime sleepiness (ESS), and all sleep diary variables, and increased PSG-based TST.
Ulmer et al., 2011 [35]	22	USA 46.0 ± 11.1 32%	Veterans with PTSD (DSM-IV-R) and insomnia disorder (DSISD, ISI > 14 and nightmares)	Individual CBT-I and IRT for nightmares plus usual care	Usual care	Subjective: ISI; PSQI; sleep diary (SOL, WASO, TST, SE, nightmare frequency)	None	Combined sleep intervention had positive effects on insomnia severity (ISI), sleep quality (PSQI), and all diary outcomes, compared with care as usual.
Gehrman et al., 2020 [36]	95	USA 55.1 ± 12.2 14%	Veterans with PTSD (DSM-IV-TR) and insomnia (ISI > 14 and sleep problems > 6 mo)	Group CBT-I via video telehealth	Group CBT-I in person	Subjective: ISI	3-mo	Based on changes in ISI scores, telehealth CBT-I was non-inferior to in-person CBT-I.
Germain et al., 2014 [37]	40	USA 38.4 ± 11.7 15%	Combat-exposed veterans, 20% PTSD (DSM-IV) and insomnia disorder (ICSD 2nd ed. and ISI > 14)	Military version of BBT-I (BBT-I-MV)	Information-only control	Subjective: ISI; PSQI	6-mo	Greater improvements in insomnia severity (ISI) and subjective sleep quality (PSQI) in BBT-I-MV than control group. Differences between response and remission rates were insignificant.
Bramoweth et al., 2020 [38]	63	USA 55.1 ± 14.4 10%	Veterans and chronic insomnia (DSM-5 and ISI ≥ 15)	BBT-I adapted for veterans and military service members	CBT-I	Subjective: ISI; PSQI; DBAS; ESS; sleep diary (SOL, NAWAKE, WASO, TST, SE, SQ)	none	Both conditions ameliorated insomnia severity (ISI), improved subjective sleep quality (PSQI), sleep-disruptive cognitions (DBAS), and various diary-based variables. Non-inferiority determination was inconclusive.
Substance use disorders								
Arnedt et al., 2011 [39]	17	USA 46.2 ± 10.1 35%	Alcohol dependence (DSM-IV) and insomnia (ISI ≥ 8, diary-based SOL and WASO ≥ 30 min for ≥3 per week for ≥ 1 month)	Individual CBT-I adapted to persons with alcohol abuse	Behavioral placebo treatment (BPT)	Subjective: ISI; sleep diary (SOL, WASO, TST, SE, SSQ)	none	Compared with the BPT control group, the adapted CGT-I showed larger improvements in insomnia severity (ISI), SE, and WASO.
Miller et al., 2021 [40]	56	USA 22.4 ± 2.7 75%	Binge drinking (>4 drinks on one occasion) in past 30 days and insomnia disorder (DSM-IV and ISI ≥ 10)	Individual BCBT-I	Sleep hygiene education	Objective: Actigraphy (SOL, WASO, TST, SE) Subjective: ISI; sleep diary (SOL, WASO, TST, SE, SSQ)	1-mo	Compared with sleep hygiene control, CGT-I significantly and persistently improved insomnia severity, objective SE, and subjective sleep quality.
Curry et al., 2004 [41]	60	Canada 43.3 ± 10.9 30%	Moderate alcohol dependence and abstinent for ≥1 mo and insomnia disorder (DSM-IV)	Individual CBT-I self-help CBT-I with telephone support	Waitlist	Objective: Actigraphy (nocturnal activity level) Subjective: ISI; PSQI; sleep diary (SOL, WASO, TST, SE, SSQ)	3-mo 6-mo	Compared with waitlist control CBT-I (both standard and self-help) significantly improved insomnia severity (ISI), subjective sleep quality (PSQI), and all sleep diary variables, except WASO.
Lichstein et al., 2013 [42]	70	USA NR 71%	ICSD Diagnosis hypnotic dependent sleep disorder DSM-IV diagnosis insomnia disorder	CBT-I plus hypnotics withdrawal	Placebo plus hypnotics withdrawal Hypnotics withdrawal only	Objective: PSG (SOL, WASO, TST, NWAKE, SE) Subjective: sleep diary (SOL, WASO, TST, NWAKE, SE)	12-mo	Compared with both control groups, CBT-I significantly shortened subjective and objective sleep latency.
Taylor et al., 2015 [43]	23	USA NR 91%	Enrolled in medication management treatment and continued insomnia symptoms reported by their psychiatrist	BCGT-I plus medication reduction module	TAU	Subjective: ISI; sleep diary (SOL, WASO, TST, SE)	None	Compared with TAU, CBT-I significantly improved insomnia severity (ISI).
Schizophrenia spectrum disorders								
Hwang et al., 2019 [44]	63	South Korea NR 35%	Schizophrenia diagnosis (DSM-5 and PSYRATS score for either delusions or hallucinations ≥2) and insomnia symptoms (ISI ≥ 15)	group-based BCBT-I plus TAU	TAU (non-random assignment)	Subjective: ISI; PSQI (TST, SOL, SE, SSQ, sleep disturbance, daytime dysfunction)	1-mo	In comparison with TAU alone, BCBT-I significantly and persistently improved all sleep variables, with the exception of sleep disturbance.
Freeman et al., 2015 [45]	50	UK NR 32%	Diagnosis of SSD (DSM-5 and PSYRATS score for either delusions or hallucinations ≥2) and insomnia (ISI ≥ 15)	Individual CBT-I adapted to SSD	TAU	Objective: actigraphy (TST) Subjective: ISI; PSQI; sleep diary (TST, SOL, WASO, SE.)	3-mo	Improvements in subjective sleep (ISI, PSQI, TST, SOL, WASO) post treatment and at follow-up. There was more ISI-based remission of insomnia in CBT-I (41%) compared with TAU (4%).
Attention deficit hyperactivity disorder								
Jernelöv et al., 2019 [46]	19	Sweden 37.0 68%	ADHD diagnosis and self-reported sleep problems	CBT-I-ADHD group intervention	None	Subjective: ISI	3-mo	ISI declined significantly from pre- to posttreatment and remained low during follow-up.
Becker et al., 2022 [47]	15	USA 14.93 ± 1.39 50%	ADHD diagnosis (DSM-5, predominantly inattentive or combined type) and sleep problems: (CMEP ≤ 23, or actigraphy-based TST < 8 or >10 h or PSQI ≥ 5)	TranS-C	None	Objective: Actigraphy (SOL, TST, SE, WASO, TIB) Subjective: PSQI, sleep diary (SOL, TST, SE, and WASO)	3-mo	Subjective sleep quality (PSQI), diary-based SOL improved, while objective time in bed increased.
Autism spectrum disorder								
Cortesi et al., 2012 [48]	120	Italy NR 17%	Diagnosis of ASD (DSM-IV) and insomnia (SOL and WASO > 30 min for ≥ 3 nights/week)	CBT-I for children (CBT-CI) CBT-CI plus melatonin	Melatonin control	Objective: Actigraphy (SOL, TST, WASO, SE) Subjective: CSHQ; sleep diary filled out by parents	none	Compared with the control group, CBT-CI significantly improved all objective sleep measures. The effect of CBT-CI combined with melatonin on objective sleep variables were larger than those of CBT-CI alone.
McCrea et al., 2020 [49]	17	USA 8.76 ± 1.99 29%	Diagnosis of ASD (DSM-5) and meeting DSM-5 criteria for insomnia (parent report)	CBT-CI adapted to ASD (CBT-CI-ASD)	none	Objective: Actigraphy (SOL, TST, WASO, SE) Subjective: Sleep diary (SOL, TST, WASO, SE)	1-mo	Compared with pretreatment measurements CBT-CI-ASD improved all subjective and all objective (except TST) sleep measures.
CBT-I in older people								
Omvik et al., 2006 [50]	48	Norway 60.8 ± 5.4 48%	Older adults (55+) and chronic primary insomnia (DMS-IV)	Individual CBT-I Zopiclone	Placebo	Objective: PSG (SOL WASO, TST, SE, TWT, N3) Subjective: sleep diary (SOL, WASO, TST, SE)	6-mo	Objective total wake time and N3 persistently improved with CBT-I compared with both placebo and Zopiclone. Effects on SE were only superior to placebo. The subjective measures improved over time equally in all groups.
Hinrichsen and Leipzig 2021 [51]	34	USA 77.2 ± 10.5 65.5%	Patients of a geriatric primary care practice with insomnia disorder (DMS-5)	CBT-I	None	Subjective: ISI; ESS; sleep diary (SOL, WASO, EMA, TST, SE)	None	Significant improvement of ISI, ESS, and diary-based SOL, WASO, EMA, and SE from pre- to posttreatment.
Buysse et al., 2011 [52]	82	USA NR 68%	Older adults with primary insomnia (DSM-IV or ICSD-2)	BBT-I	Information control	Objective: PSG and actigraphy (both SOL, WASO, TST, SE) Subjective: PSQI; sleep diary (SOL, WASO, TST, SE)	6-mo	BBT-I produced larger improvements in actigraphy-based SOL, WASO, and SE and in all diary-based sleep variables compared with the control group.
McCrae et al., 2018 [53]	62	USA 69.5 ± 7.7 42%	Older adults (65+) with chronic insomnia complaints (SOL or awake during night > 30 min, ≥3 nights/week, for ≥6 mo)	Individual BBT-I	Social conversation training	Objective: Actigraphy (SOL, WASO, TST, and SE) Subjective: sleep diary (SOL, WASO, TWT, TST, SE)	3- mo	Significant and persistent improvements in subjective, but not objective SOL, WASO, and SE compared with the control group.
McCurry et al., 2021 [54]	327	USA 70.2 ± 6.8 74.6%	Older adults (60+) with insomnia symptoms (ISI ≥ 11) and osteoarthritis-related pain symptoms	Telephone CBT-I	Education only control	Subjective: ISI	12-mo	Telephone CBT-I significantly and persistently improved ISI scores compared with control.
Sadler et al., 2018 [55]	72	Australia NR 56%	Older adults (65+) with insomnia disorder (DSM-5) and comorbid MDD (DSM-5)	Standard CBT-I Advanced CBT-I	Psychoeducation control	Subjective: ISI; sleep diary (SOL, WASO, TST, SE)	3-mo	The standard and advanced CBT-I groups had both significantly and persistently better subjective sleep outcomes than the control group.
Cassidy-Eagle et al., 2018 [56]	28	USA 89.4 85.7%	Older adults in residential care facilities for the elderly with insomnia disorder (DSM-IV) and MCI.	Adapted CBT-I group intervention	Active control group	Objective: Actigraphy (SOL, WASO, TST, and SE) Subjective: ISI	4-mo	Actigraphy-based SOL, WASO, and SE and ISI score were significantly improved in the treatment group compared with the control group.

Abbreviations: ADHD—attention deficit hyperactivity disorder; ASD—autism spectrum disorder; BBT-I—brief behavioral treatment for insomnia; BBT-I-MV—BBT-I-military version; BCBT-I—brief cognitive behavioral treatment for insomnia; BDI—Beck depression inventory; BT—behavioral treatment; BT-I—behavioral therapy for insomnia; CBT-CI—cognitive behavioral treatment for childhood insomnia; CBT-CI-ASD—cognitive behavioral treatment for childhood insomnia in autism spectrum disorder; CBT-D—cognitive behavioral treatment for depression; CBT-I—cognitive behavioral treatment for insomnia; CDRS-R—children’s depression rating scale-revised; CMEP—Children’s morningness–eveningness preferences scale; CSHQ—Children’s Sleep Habits Questionnaire; CT—cognitive treatment; CT-I—cognitive therapy for insomnia; DBAS—dysfunctional beliefs and attitudes about sleep; DSISD—Duke structured interview for sleep disorders; EMA—early morning awakening; ESS—Epworth sleepiness scale; GAD—generalized anxiety disorder; GRID-HAMD;—GRID–Hamilton rating scale for depression; HAMD17—Hamilton rating scale for depression; HDRS—Hamilton Depression Rating Scale; HRSD_17_—Hamilton rating scale for depression, 17 items; ICBT-I—internet-delivered cognitive behavioral treatment for insomnia; ICSD—international classification of sleep disorders; IDS-C—inventory of depressive symptomatology; IRT—imagery rehearsal therapy; ISI—Insomnia severity index; MBCT-I—mindfulness-based cognitive therapy for insomnia; MCI—mild cognitive impairment; MDD—major depressive disorder; NAWAKE—number of awakenings; NR—not reported; OCD—obsessive compulsive disorder; PSYRATS—Psychotic Symptom Rating Scales; PSQI—Pittsburgh sleep quality index; PSG—polysomnography; PHQ-9—patient health questionnaire-9; SE—sleep efficiency; SM—sleep maintenance; SOL—sleep onset latency; SSS—Stanford sleepiness scale; SQ—sleep quality; SSQ—subjective sleep quality; TAU—treatment as usual; TransS-C—trans diagnostic sleep and circadian; TST—total sleep time; TWT—total wake time; WASO—wake after sleep onset; YMRS—Young mania rating scale.

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
