# Peer review of "Effectivity of (Personalized) Cognitive Behavioral Therapy for Insomnia in Mental Health Populations and the Elderly: An Overview"

_jpm, 2022, doi:10.3390/jpm12071070_

Round 1

Reviewer 1 Report

In the present narrative review the Authors aimed to describe existing modifications of CBT-I as well as their efficacy in specific mental health populations. First, healthy sleep, sleep assessment, insomnia disorder and the most common insomnia treatments are discussed. Hereafter, a description is given of CBT-I and related non-pharmacological insomnia therapies for the following psychiatric disorders: major depressive disorder (MDD), bipolar disorders, anxiety disorders, posttraumatic stress disorders (PTSD), substance use disorders (SUD), schizophrenia spectrum disorders (SSD), attention deficit hyperactivity disorder (ADHD) and autism spectrum disorders (ASD). Additionally specific attention was given by the Authors to insomnia in elderly persons. Finally, the Authors provided a overarching conclusion, gaps in literature and future perspectives.

Overall, I found this study timely, original, well conducted and scientifically sound. I have some suggestions aimed to improve the quality of the paper and these are outlined below:

1) In the introduction a brief note on the effect of insomnia on the potential development of suicide ideation per se independently from the presence of pasychiatric disorders should be added with appropriate references (see doi 10.3390/brainsci11121586). This was a phenomenon most frequent seen in Healthcare Workers during the pandemic period.

2) Moreover, a brief note on some drugs, such as agomelatine, that might help to improve insomnia especially in MDD or bipolar depression should be added with appropriate references (see doi 10.2147/NDT.S41557).

3) In addition a Table depicting the major sleep alteration in psychiatric disorders (especially MDD) should be useful to the reader.

Author Response

We thank reviewer#1 for his/her positive evaluation of the manuscript.

  • In the introduction a brief note on the effect of insomnia on the potential development of suicide ideation per se independently from the presence of psychiatric disorders should be added with appropriate references.

A very good suggestion. We added to Introduction that insomnia increases the risk for suicidal thoughts and behavior and referred to a recent study of Bishop et al. 2020 convincingly showing this.

  • Moreover, a brief note on some drugs, such as agomelatine, that might help to improve insomnia especially in MDD or bipolar depression should be added with appropriate references (see doi 10.2147/NDT.S41557).

Thank you for this comment. Unfortunately, the evidence for efficacy of agomelatine in the treatment of insomnia is scarce and weak. In the paper suggested by the reviewer insomnia was not a selection criterion for the participants of the study. Therefor we decided not to include this particular paper. However, we did include a review of Atkin et al. 2017 that provides a broader overview of the pharmacological treatment of insomnia.

  • In addition a Table depicting the major sleep alteration in psychiatric disorders (especially MDD) should be useful to the reader.

Insomnia symptoms are transdiagnostic symptoms of mental disorders and are not fully distinguishable between different specific mental disorders (see Baglioni et al. Sleep and mental disorders: a meta-analysis of polysomnographic research, Psychol Bull 2016, 142(9):969-990). Therefore, it is not informative to present such a table. Furthermore, we feel that such a table would fall beyond the scope of our manuscript.

Reviewer 2 Report

Authors have done a narrative review on Effectivity of (personalized) cognitive behavioral therapy for insomnia in mental health populations and elderly. Introduction is appropriate, highlights the importance of insomnia and comorbid mental health issues. Methods are well described. Authors have done a good workin identifying and reporting studies where CBT-I has been modified for comorbid psychiatric conditions and have discussed conclusion at the end of each section, which is helpful for the readers. Discussion is rich in content and appropriate, they have further discussed the reasons why larger RCT are limited and possible confirmation biases as well. Gaps in literature are to the point and encompass major issues They have also proposed a unique solution to improve the database and study different versions of CBT under one international database. This, although unique and tempting can have its own logistical challenges, which should have also been discussed.

Please add challenges of the proposal of international database.

Author Response

We thank reviewer#2 for his/her very positive evaluation of the manuscript.

  • This, although unique and tempting can have its own logistical challenges, which should have also been discussed. Please add challenges of the proposal of international database.

We agree with the reviewer and have added the following sentence to the Perspective: Naturally, this proposal comes with its own challenges such as heterogeneity in assessed populations and languages, problems in quality control like potential protocol misalignments between different mental health care institutes, and the burden of structural database maintenance (e.g. financial, time and staff).